# Methodological Issues in the Study of the Development of Pain Responsivity in Preterm Neonates: A Systematic Review

**DOI:** 10.3390/ijerph17103507

**Published:** 2020-05-17

**Authors:** Damiano Menin, Marco Dondi

**Affiliations:** Department of Human Studies, University of Ferrara, via Paradiso, 12, 44100 Ferrara, Italy; damiano.menin@unife.it

**Keywords:** pain ontogeny, premature birth, gestational age, postmenstrual age, chronological age

## Abstract

The study of the development of neonatal pain responses is of key importance, both for research and for clinical reasons, with particular regard to the population of preterm neonates, given the amount of painful procedures they are exposed to on a daily basis. The aim of this work was to systematize our knowledge about the development of pain responses in prematurely born neonates by focusing on some key methodological issues. Studies on the impact of age variables, namely gestational age (GA), postmenstrual age (PMA) and chronological age (CH), on pain responsivity in premature neonates were identified using Medline and Scopus. Studies (*N* = 42) were categorized based on terminological and methodological approaches towards age variables, and according to output variables considered (facial, nonfacial behavioral, physiological). Distinct multidimensional developmental patterns were found for each age-sampling strategy. Overall, each of the three age variables seems to affect pain responsivity, possibly differently across age windows. Targeted as well as integrated approaches, together with a renewed attention for methodological consistency, are needed to further our knowledge on this topic.

## 1. Introduction

The early ontogeny of pain reactivity is a key theme for the study of the emergence of human consciousness and emotions [1,2]. The abilities to perceive pain and mount a multidimensional response to a noxious stimulus, in fact, have been deemed as crucial steps in the constitution of a minimal level of awareness [3].

The rising interest towards pain assessment and management in preterm newborns is also driven by clinical considerations, since the same clinical practices that now provide increasing survival rates even for neonates born at extremely low gestational ages (less than 28 weeks) result, in fact, in an average of approximately 15 stressful procedures per day [4,5,6]. In particular, the well-founded concerns for adverse effects of early pain exposure [4,5,7,8,9,10,11,12,13] have contributed to increase the attention towards neonatal pain and preterm birth.

The variability known to characterize pain ontogeny during the perinatal period [14,15,16,17] makes crucial the issue of understanding its early dynamics. Given their peculiar status, the development of preterm neonates (i.e., those born before 37 completed weeks of gestation) can be framed in terms of the interaction between three separate chronological variables, i.e., the duration of the gestation (gestational age, GA), the time from birth to observation (chronological age, GA) and the sum of the two measures above (postmenstrual age, PMA). Understanding their separate and joint effects would enhance our diagnostic ability in recognizing atypical performances, allowing clinicians to adequately weight different age variables for pain assessment and management. Moreover, the study of age-related variations of pain reactivity in preterm neonates can provide important insights into the early ontogeny of pain, since this analysis allows us to inquire into the separate effects of intrauterine (GA), general (PMA) and extrauterine (CH) development and experiences (in particular, with regard to pain exposure in the Neonatal Intensive Care Unit, NICU).

### 1.1. Age Terminology

A preliminary issue that needs to be addressed concerns nomenclature and methods for age calculation. The recent proliferation of works focused on preterm birth in many different domains of clinical as well as interdisciplinary research has probably contributed to the spread of a variety of nonstandardized nomenclatures to label age measures. In order to reduce ambiguities and promote a standardized terminology, a policy statement has been issued by the American Academy of Pediatrics (AAP) [18] defining gestational age (GA) as the number of completed weeks elapsed between the first day of the last menstrual period and the day of delivery, chronological age (CH, also postnatal age, PNA), as the time elapsed between birth and observation, and postmenstrual age (PMA) as the sum of gestational age and chronological age. The use of “conceptional age”, which refers to the time elapsed between conception and day of delivery and postconceptional age (PCA), defined as the time elapsed between conception and observation, is discouraged within this policy statement in order to avoid confusion.

These definitions highlight a controversial point, as “postmenstrual” and “postconceptional age” on one hand, and “gestational” and “conceptional age” on the other, are often used as synonyms, hence the suggestion to not adopt conception-related terms.

### 1.2. GA Assessment Methods

The spread of obstetric ultrasound as a mean to estimate the date of conception raises some problems regarding the consistency of the measure of GA, as highlighted by a brief characterization of the most commonly employed methods.

The last menstrual period-based (LMP-based) method consists in starting counting GA from the first day of the last normal menstrual period. Despite its tendency to overestimate the length of gestation by approximately two weeks [19], it is still the most common measure of GA, due to its almost universal availability. However, its reliability is conditioned to individual cycle length and the precision in recalling the exact date.

Ultrasound-based (US-based) estimates, on the other hand, are usually carried out before 20 weeks of gestation, adopting the fetal crown-rump length. This method is known to be more precise than LMP-based measures [20]. However, it relies on the assumption of typical development, making problematic its adoption with atypically developing populations. In fact, US-based gestational age assessment has been proven prone to systematic errors in the case of congenital malformations [21], maternal obesity [22] or severe delays in intrauterine growth [23]. Moreover, US-based estimates do not measure GA, but rather conceptional age so they are conventionally added with two weeks to obtain GA, as growth charts have been constructed using LMP-based measures [24]. Finally, postnatal assessment is characterized by scarce precision, and is performed mainly in order to establish the actual developmental level of a neonate [25].

### 1.3. Outcome Measures

Another issue affecting the effort to disentangle the effect of age variables on neonatal pain reactivity concerns the methodologies adopted in order to measure pain responses. Interstudy comparability is currently limited due to the adoption of different algometric methods, including physiological indexes, observational scales for the analysis of facial or nonfacial behavior, and multidimensional scales combining physiological and behavioral indexes. 

### 1.4. Age-Sampling Strategies 

Since various studies highlighted distinct effects of PMA [16,26], GA [27,28] and CH [29,30] on neonatal pain responsivity, the question about developmental paths in preterm pain responses should be understood in terms of the relations existing between each age measure—i.e., GA, CH and PMA—and different physiological and behavioral pain-related indexes. Age measures, however, cannot be considered as autonomous variables, since PMA is defined as equal to the sum of GA and CH. It is therefore impossible to contrast two groups differing only for one of these variables, in order to isolate its effect on pain responses. Any study using a sample homogeneous for one age variable, either GA, CH or PMA, would be in fact subject to almost perfect collinearity between the remaining two [31,32,33]. 

We can therefore distinguish (a) studies recruiting subject at fixed PMA, (b) studies recruiting subject at fixed GA, (c) studies recruiting subject at fixed CH and (d) studies adopting alternative strategies, without keeping any age variable fixed. Each of these strategies allows us to inquire a peculiar empirical question regarding the early development of pain responses. 

### 1.5. Aims

As developmentalists, our main interest regarding the study of neonatal pain lies in disentangling the existing relations between age measures and pain responses, in order to increase our knowledge about the very early development of pain. In the present paper, we will pursue this core aim by performing a systematic review of the literature on the effects of age variables towards pain responses in preterm neonates. Studies will be categorized based on different approaches towards the issues introduced in the previous paragraphs, namely (a) age-terminology, (b) GA assessment methods, (c) outcome measures and (d) age-sampling strategies.

The central focus of the current review was on age-sampling variables, because this issue concerns directly the approaches adopted to study the relation between age variables (i.e., GA, CH, PMA) and pain responsivity in preterm neonates. In particular, we will pinpoint outlooks and inescapable limits of each age-sampling strategy, focusing on how they shape the nature of the empirical questions that have been asked in investigating the relation between age variables and pain responses in preterm neonates. In light of this analysis, we will be able to assess the state of the art concerning the effort to disentangle the effects of age variables on pain responsivity in preterm neonates, to identify issues hampering interstudy comparability and to suggest directions for future research.

## 2. Methods

### 2.1. Eligibility Criteria

We selected studies for review based on the following eligibility criteria—(1) the study was a full-report written in English language and published in a peer reviewed journal; (2) the study included at least 20 preterm neonates within its participants (born before 37 completed weeks of gestation); (3) the study included statistical analyses aimed at investigating age-related differences in pain responsivity; (4) the study included, as outcome measures, at least one physiological, behavioral or multidimensional index of pain responsivity; (5) the demographic information provided included data about at least two age variables (among GA, CH and PMA).

### 2.2. Search Strategy

The literature search was performed in April 2020. Studies up to March, 2020 were identified through a search of Medline and Scopus. In order to include all potential studies investigating preterm neonates, the search was set up to include all records mentioning at least one keyword among “neonate”, “newborn” and “infant” together with “premature” or “preterm”. Finally, both “pain” and “age” were included in the search filter. Upon reading the abstracts, articles deemed to meet the inclusion criteria were read in full.

### 2.3. Data Extraction

In order to compare sample demographics from different studies, we adopted the standard terminology proposed in the AAP guidelines. Terminological choices and methods for age calculation adopted have been reported in the results section, together with a survey of outcome measures. All the papers collected were subsequently categorized into four groups, based on the age-sampling strategy they adopted. We distinguished between studies keeping PMA fixed (Group A), studies keeping GA fixed (Group B), studies keeping CH fixed (Group C) and other studies (Group D). The studies combining more than one approach were included in all of the pertinent groups.

Because of the heterogeneity observed among studies concerning these methodological issues, a meta-analysis was deemed inappropriate. A qualitative review was therefore performed following the Preferred Reporting Items for Systematic Reviews and Meta-Analyses (PRISMA) guidelines. 

## 3. Results

After removing duplicates and adding eligible studies from reference lists, a total of 1457 articles were screened. Full-text assessment was carried out for 148 studies. Figure 1 shows the PRISMA Flowchart for the literature search.

Only 42 studies were found to meet the selection criteria and were assigned to one or more non-mutually-exclusive groups, as showed in Table 1. 

Thirteen were included in Group A (PMA fixed) [15,16,27,28,31,34,35,36,37,38,39,40,41], 11 in Group B (GA fixed) [14,15,16,30,40,42,43,44,45,46,47], 17 in Group C (CH fixed) [14,15,16,29,30,40,48,49,50,51,52,53,54,55,56,57,58] and ten in Group D (none fixed) [26,29,59,60,61,62,63,64,65,66].

### 3.1. Age Terminology

To designate the sum of GA and CH (PMA), 21 out of 42 studies adopted either the term “postconceptional age” [15,31,34,35,36,37,38,39,40,41,43,44,50] or a variant such as “postconceptual” [27,48,55,60,64], “conceptual” [16,59] or “postconception age” [45]. Additionally, eight studies adopted the label “gestational age” to designate both GA and the sum of GA and CH (PMA) [14,42,46,56,57,58,65,66], and nine studies did not offer any terminological specification for PMA [29,47,49,51,52,53,54,61,62]. Only four studies [26,28,30,63] adopted the terminology “postmenstrual age”, as suggested in the AAP policy statement [18]. Furthermore, to designate the time elapsed from birth, only three studies adopted the suggested terminology “chronological age” [14,43,56], whereas 24 studies used the expression “postnatal age” [14,16,26,27,28,29,35,36,37,38,40,44,45,49,51,53,55,59,60,62,63,64,65,66] and 17 did not adopt any specific expression to designate this measure [15,29,30,31,34,39,41,42,46,47,50,51,52,53,54,61,62].

### 3.2. GA Assessment Methods

Thirty three out of forty-two studies did not explicitly provide information about the methods used to assess GA [14,15,16,27,29,30,31,34,35,36,37,39,41,42,43,44,45,46,47,48,49,50,51,52,53,56,57,58,60,61,62,64,66]. Eight studies established GA based on US when available [26,28,38,54,55,59,63,65], otherwise adopting LMP-based estimates [26,28,54,55,63,65] or a combination of LMP, US examinations later in pregnancy and postnatal assessment [38,59]. One study calculated GA as the average between US- and LMP-based estimates [40]. In order to compare and discuss results from these studies, we assumed that all US-based measures presented in the studies we examined were adjusted in order to be consistent with the LMP-based method, according to which a full-term gestation lasts between 38 and 42 weeks [20].

### 3.3. Outcome Measures

Out of the ten algometric scales adopted by at least one study, five, including the Neonatal Facial Coding System (NFCS) [67], the Newborn Individualized Developmental Care and Assessment Program (NIDCAP) [68], the Face, Legs, Activity, Crying, Consolability scale (FLACC) [69], the Echelle Douleur Inconfort Nouveau-Né (EDIN) [70] and the Behavioral Indicators of Infant Pain scale (BIIP) [71,72] are unidimensional scales, composed of items relative to facial behavior (NFCS, EDIN, FLACC, NIDCAP, BIIP), nonfacial behavior (EDIN, FLACC, NIDCAP, BIIP) and information relative to consolability or other interactional features (EDIN, FLACC). The remaining five scales, including the Premature Infant Pain Profile (PIPP) [73], the Neonatal Infant Pain Scale (NIPS) [74], the CRIES Scale (Cry, Requires O_2_, Increased vital signs, Expression, Sleeplessness) [75], the Neonatal Pain, Agitation and Sedation Scale (N-PASS) [76] and the Bernese Pain Scale for Neonates (BPSN) [77] are multidimensional tools, combining behavioral and physiological items, e.g., blood pressure (N-PASS), heart rate variability (N-PASS, PIPP, BPSN), oxygen saturation (N-PASS, CRIES, PIPP, BPSN) or respiratory rate (N-PASS, NIPS). 

Twenty-seven studies adopted at least one of these algometric scale as outcome measure, namely NFCS [27,31,34,35,36,37,38,41,45,49,53,54,55,60], NIDCAP [35,36,37,41,44], EDIN [62], BIIP [40] or FLACC [52] among unidimensional scales and PIPP [14,44,45,46,50,51,52,58,64], NIPS [30,56], CRIES [52] or N-PASS [47] among multidimensional scales. Nine studies, instead of an existing algometric scale, adopted various behavioral indexes as dependent variables, pertaining to facial [42,43,48,49,53,54,55,66] or nonfacial [15,48,49,53,54] cues, while three studies used as an outcome measure of behavioral states [16] and acoustic features of neonatal cries [53,55], respectively. Physiological outcomes were adopted by 32 studies, namely heart rate variability [16,27,29,37,41,42,44,45,48,49,53,54,55,57,60], oxygen saturation [16,27,29,37,41,42,44,45,48,53,54,55,57,60], plasma [34,39,54] or salivary [39,51,61] cortisol levels, skin conductance [47,59], respiratory rate [44], blood pressure [16], intracranial pressure [60], adrenocorticotropic hormone [39] or 17-hydroxyprogesterone [61] levels. Six studies adopted neuroimaging techniques, including electroencephalography (EEG) [28,40,65,66] and near-infrared spectroscopy (NIRS) [26,29].

### 3.4. Age-Sampling Strategies

#### 3.4.1. Group A. Studies Recruiting Subjects at Fixed PMA

A first strategy consisted in adopting a sample relatively homogeneous with respect to PMA, contrasting—either within the same group or between different groups—neonates born at different GAs, correlatively observed at different CHs.

Thirteen studies adopted this strategy. Eleven of them recruited participants averaging between 32 and 33 weeks PMA [15,27,31,34,35,36,37,38,39,40,41], while one included neonates at 36 weeks PMA [16] and one compared preterm neonates at term-equivalent age with full term newborns [28]. Six of these studies [34,36,37,38,39] used a crossover design, comparing responses to variously stressful procedures [16,34,37], clustered care (CC) versus rest [39], pain following CC versus pain following a rest period [38] or CC following rest versus CC following a painful procedure [36]. The remaining studies tested the responses to a singular painful procedure, consisting of a heel lance for blood sampling [15,27,28,31,35,40,41].

Ten studies used behavioral outcome measures, namely NFCS [27,31,34,35,36,37,38,41], BIIP [40] or NIDCAP [35,36,37,41]. One study included coding of behavioral states [16]. Eleven used physiological outcome measures—heart rate variability [15,16,27,31,35,37,38,41], plasma cortisol [34,39], adrenocorticotropic hormone [34,39], oxygen saturation [15,16,27,37,41], EEG [40] and evoked potentials [28]. Multidimensional algometric tools were not adopted by any study from this group.

Aside from GA, which was the main independent variable in each study, the most controlled predictor variable was the total number of skin-breaking procedures undergone during the last 24 h [39] or from birth [27,31,34,35,36,37,39,41]. Moreover, three studies tested the effect of morphine exposure since birth [31,34,35]. Notably, the impact of CH was not explicitly inquired.

The overall tendency associates lower GA with dampened facial expressions [27,31,38,41], heightened nonfacial behavioral [35,37,41] and physiological [15,27,28,31] responses. Interestingly, a study found a nonlinear relation between GA and the magnitude of changes in heart rate following painful procedures [16]. In particular, neonates between 28 and 32 weeks GA and neonates older than 36 weeks GA exhibited respectively the largest and the weakest HR variations, while neonates younger than 28 weeks GA and between 32 and 36 weeks GA showed intermediate responsivity.

#### 3.4.2. Group B. Studies Recruiting Subjects at Fixed GA

A second way that has been employed to address the relation between age measures and pain responses consisted of adopting a sample homogeneous for GA, typically by performing longitudinal or repeated-measures inquiries. Despite including also between-subject elements, mostly comparing different GA groups [14,15,16,47], the main purpose of these studies remains to investigate the evolution of pain responsivity within subjects over time, inquiring about the effect of concurrently increasing PMA and CH. Since, and to the extent to which, the within-subject component is predominant, we can consider GA as a fixed variable in these studies.

Out of the 11 studies adopting this strategy, nine performed longitudinal inquires on neonates born at GAs ranging from 22 to 37 weeks [14,15,16,30,42,43,45,47] and, with one exception [44], including at least one observation during the first week of postnatal life. Two studies were limited exclusively to this period [16,45]. One study [40] compared two distinct groups of neonates born at the same GA (averaging 29 weeks), observed respectively at 30 and 33 weeks PMA, while one was a repeated-measure design [46] including two observations, respectively, at 33 and 35 weeks PMA. Two studies used a crossover design, contrasting responses to real vs. sham heel lance [42] or to various invasive routine procedures [16] performing multiple observations; nine tested the responses to a singular painful procedure, consisting of heel lance [14,15,16,30,43,44,47], venipuncture [45] or retinopathy of prematurity screening [46].

Eight studies used behavioral outcome measures based on tools for analysis of facial [42,43,45] and nonfacial behavioral cues [15,16,40,42,44,45]; nine studies used physiological outcome measures, namely heart rate variability [15,16,42,43,44,45], oxygen saturation [15,16,42,44,45], respiratory rate [44], skin conductance [47] and EEG [40]. Six studies used multidimensional assessing tools [14,30,44,45,46,47].

The overall findings from these studies indicate a general increase of behavioral responses in neonates studied longitudinally, including measures of facial [42] and nonfacial behavior [30,44], and a similar increase in physiological pain responses [16,42] and NIPS scores [30] over time, while PIPP scores were found to decrease [46] and the remaining four studies employing multidimensional outcomes did not highlight any longitudinal change [14,44,45,47].

#### 3.4.3. Group C. Studies Recruiting Subject at Fixed CH

A third option consisted of adopting a sample homogeneous for CH, hence testing primarily the concurrent effect of increasing GA and PMA. Seventeen studies adopted this strategy. Thirteen of them included preterm newborns within their first week of life [14,15,16,29,30,48,50,52,54,57], or within the first two weeks [49,56,58]. One [40] compared two groups of neonates approximatively one week after birth. Another study performed two observations, the first between postnatal days 2–7 and the second between postnatal days 10–18 [51], on preterm and full-term neonates. The last two studies, studying neonates within the first six weeks of postnatal life, and contrasting neonates born at different GAs [53,55] were included in this group despite the higher variability in CH compared to other studies, because their analyses did not control for the effects of CH or PMA.

Four studies performed multiple observations contrasting various invasive procedures in a crossover design [16,52,54,57], thirteen studies investigated the responses to a singular procedure, consisting of a heel lance [14,15,30,40,48,49,50,53,55,58], venipuncture [29], electrocardiogram (ECG) [56] or diaper change [51].

The behavioral outcomes employed were FLACC [55], NFCS [49] behavioral states [16] and selections of facial [48,53,54,55] and nonfacial [15,48,53,54] behavioral categories. Additionally, one study performed an analysis of acoustic features of cries [53]. The physiological measures and tools adopted were oxygen saturation and heart rate variability [16,29,48,53,54,55,57], heart rate variability alone [15,49], skin conductance [51], blood pressure [16], cortisol [51,54], HbO2 increases in the somatosensory cortex, recorded by NIRS [29] and EEG [40]. Seven studies adopted multidimensional tools, namely PIPP or PIPP-R [14,50,51,52,58], NIPS [30,51,56], and CRIES [52].

Results from these studies seem to associate lower GAs with dampened facial [48,49,55,58] and nonfacial [30,48,58] behavioral reactivity, as well as with stronger physiological responses [15,29,58], slower return to baseline physiological values [57] and weaker pain-related increases in BIIP [40], CRIES and FLACC scores [52]. Notably, one study [14], considering a group of newborns between 24 and 36 weeks GA, found that neonates born between 31 and 33 weeks GA exhibited the larger PIPP scores following heel lance, highlighting a nonlinear relation between GA and pain responses. Furthermore, two studies, found GA to be negatively associated with variations in PIPP [51] and NIPS [51,56] scores during mildly stressful procedures, while another study found an opposite positive effect of GA on PIPP scores [50] during heel lance.

#### 3.4.4. Group D. Alternative Strategies

Despite covering the vast majority of the existing literature, the approaches introduced above did not represent the only way to investigate age-related changes in preterm neonates. In fact, two alternative sampling strategies have been adopted in order to isolate the effect of a single age variable on pain responses, overcoming the collinearity issue constitutively affecting the approaches presented in the previous paragraphs.

##### Group D1. Studies Inquiring Limited Age-Windows

A first alternative strategy is exemplified by studies recruiting newborns within the very first postnatal days [29,61], in order to disentangle the effect of CH on pain responses. The two studies adopting this approach employed exclusively physiological indexes of pain responsivity, including heart rate variability, oxygen saturation, pain-induced increases of HbO2 [29], salivary cortisol and 17-hydroxyprogesterone levels [61]. Results highlighted a positive association between CH and pain-related increases of HbO2 [29] and a negative association between CH and baseline salivary cortisol levels [61].

##### Group D2. Studies with Heterogeneous Samples

A second strategy potentially overcoming the collinearity issue consists of adopting a sample characterized by a wide range for GA, CH and PMA together [26,59,60,62,63,64,65,66]. The behavioral tools employed were NFCS [60], the EDIN scale [62], facial grimacing [66] and the latency to facial response [63]. The physiological measures adopted were changes in hemoglobin concentration detected via NIRS [26], skin conductance [59], EEG measures [65,66], intracranial pressure, heart rate variability and oxygen saturation variations [60]. One study used PIPP scores [64]. The results from studies adopting this approach associate lower GAs with delayed facial reactivity [63] and lower EDIN scores [62], and lower PMAs with dampened facial [60,66] and physiological [59,60] pain responsivity. Furthermore, three studies found similar results associating a higher PMA with stronger and more specific pain responses measured respectively via NIRS [26] and EEG [65,66]. One study reported opposite results, associating lower PMAs with dampened multidimensional PIPP scores [64], while highlighting no effects of CH.

## 4. Discussion

### 4.1. Age Terminology, GA Assessment Methods, and Outcome Measures

We found interstudy comparability to be hampered by some fundamental problems concerning age terminology, GA assessment methods and outcome measures adopted. Six different terms have been used to designate the sum of GA and CH. Only four, out of forty-two studies considered, adopted the label “postmenstrual age” suggested in the AAP guidelines [26,28,30,63]. Furthermore, only three studies referred to “chronological age” [14,43,56].

Moreover, 33 out of 42 studies matching the inclusion criteria did not report the methods adopted in order to assess and eventually adjust GA, while the remaining nine highlighted a significant variability in the techniques adopted. Five behavioral and five multidimensional scales, together with several physiological and neurophysiological indexes, were adopted by at least one of the forty-two studies that matched selection criteria.

### 4.2. Age Sampling Strategies

More importantly, the above survey confirms the crucial role played by the selection of different age-sampling strategies in shaping the research questions concerning the early development of pain responses, as well as the results obtained.

Results from Group A (studies recruiting subjects at fixed PMA) found lower GA to be associated with dampened facial, increased physiological and nonfacial behavioral responsivity. Studies from Group C (recruiting subjects at fixed CH), on the other hand, found that more premature newborns were exhibited dampened behavioral (both facial and nonfacial) responsivity, together with increased physiological responsivity in the first postnatal week. Longitudinal studies (Group B, recruiting subjects at fixed GA) have shown an increase in physiological, facial and nonfacial behavioral pain responses over time. Finally, limited data from Group D1 (studies inquiring about limited age windows) highlighted a CH-related increase in physiological responsivity in newborns, as well as a CH-related decrease in baseline cortisol levels, suggesting a stabilization of pain responsivity in the very first postnatal days. Studies from Group D2 (studies with heterogeneous samples) found physiological and facial reactivity to be positively associated with PMA.

### 4.3. General Discussion

Overall, the above survey highlighted a high degree of heterogeneity across the considered literature, regarding both the very strategies adopted to study developmental changes in pain responsivity of preterm neonates, and some more specific methodological options concerning age terminology, GA assessment and outcome (pain) measures. Each of the age-sampling strategies we reviewed has proven insufficient, at least taken alone, to disentangle the existing relations between age variables (i.e., GA, PMA, CH) and multidimensional patterns of pain responses. In particular, the strategies adopted by Groups A, B and C are each meant to satisfy the demand for homogeneity regarding one age variable (respectively, PMA, GA and CH). However, because of the collinearity between the remaining two variables these approaches entail, they are ultimately unable to clarify the dynamics of intra- and extrauterine development of pain responses. Alternative approaches (Group D), on the other hand, can in principle overcome these collinearity issues, allowing the isolation of the effect of a specific age variable on pain responsivity, and could therefore be identified as the best way to investigate the development of pain responsivity in preterm neonates. However, they too present significant limitations. In particular, the approach of inquiring about limited age windows (Group D1) [29,61] allows us to disentangle the effects of small variations in CH on pain responses, but it is only viable when we expect to find dramatic changes within a very small age-window, as in the case of the first postnatal days. On the other hand, adopting an heterogeneous sample (Group D2) can allow us to disentangle the effect of separate age variables on pain responses across the whole preterm period. However, this approach is particularly demanding in terms of sample size and complexity of the required analyses, as the represented population is more heterogeneous than required in other strategies. Moreover, especially since some studies showed a nonlinear relation between age variables and pain measures [14,16], inferences based on limited age-windows cannot be used to establish general tendencies. This issue is well-illustrated by the fact that eight [27,31,34,35,36,37,38,39] out of thirteen studies from Group A (keeping PMA fixed), e.g., included only neonates between 31 and 34 weeks PMA.

Other methodological issues were found to preclude a fine-grained depiction of patterns of developmental change for neonatal pain, namely regarding the definition of predictors (age terminology and GA assessment methods) and outcomes (i.e., pain measures) of such analysis. In particular, the spread of various age-related terminological options highlighted a critical heterogeneity in the nomenclature and definition of age variables. This situation is complicated by the variability and general lack of data regarding methods adopted to assess GA. In fact, most studies adopted cut-off values in discrepancies between different estimates in order to identify implausible gestational ages, despite the availability of more elaborated approaches, based e.g., on mixture models [78,79,80,81]. For these reasons, full disclosure of measures and tools used to date pregnancies, as well as the adoption of a shared terminology, are needed in order to establish the basis for any rigorous effort in studying the development in the preterm period. Moreover, the adoption of sophisticated and elastic tools like the Bayesian mixture model developed by Zhang and colleagues [81] could greatly contribute to data cleanliness, therefore limiting artifacts and improving reliability in age-related inquiries involving preterm neonates.

The methodological variability concerning outcome measures represents another threat to interstudy comparability, as different algometric tools hold irreconcilable assumptions about neonatal pain and its manifestations. Literature survey, in particular, raises some questions about the adoption of (a) scales including age-related items (e.g., PIPP) and (b) multidimensional scales (e.g., NIPS, CONFORT, CRIES, N-PASS, NIDCAP) or scales combining in the same score facial and nonfacial behavioral indexes (e.g., FLACC, BIIP). The former case is problematic because, as age variables are the principal predictors in these studies, their inclusion in the calculation of the main dependent variable (i.e., pain responsivity) may lead to statistical artifacts. The second issue has become apparent in light of the discrepancy highlighted among groups in age-related variations of physiological, facial and nonfacial behavioral indexes. These considerations might explain, e.g., the fact that, out of six longitudinal studies adopting multidimensional scales, two highlighted, respectively, an increase in NIPS scores [30] and a decrease in PIPP scores [46] over time, while the remaining four studies did not find significant age effects [14,44,45,47].

## 5. Conclusions

Despite all the issues and limitations highlighted, a general trend emerged across different approaches, albeit based on limited evidence, a GA- and PMA-related increase in facial pain responses, with more immature neonates showing dampened responsivity, was seen. This tendency is consistent with the common opinion according to which, the older the infant, the more robust and recognizable the response [42]. However, physiological as well as nonfacial behavioral indexes did not show any consistent trend across groups and predictors considered. Overall, the concurrent variability of methodological options adopted on different levels (i.e., regarding terminology, GA assessment methods, pain measures and age-sampling strategies) determines serious problems in any attempt to compare results from different studies and attain a rigorous synthesis of our current knowledge about this extremely important topic.

In order to effectively further our understanding of the neonatal development of pain responsivity, future research should pursue a common methodological ground regarding the issues affecting this effort. In particular, based on the findings of the current review, our opinion is that this should include a shared terminology based on the AAP guidelines, the full disclosure of methods used to assess GA, the adoption of separate one-dimensional scales for physiological, facial and nonfacial behavioral responses, and a use of age sampling strategies in line with the specific research aims. Despite the complexity of the problems involved (including those regarding the relations and interactions between age measures and other, nonchronological variables) a thorough discussion of these issues is the only way to ensure a quick increase in our knowledge of this delicate topic.

## Figures and Tables

**Figure 1 ijerph-17-03507-f001:**
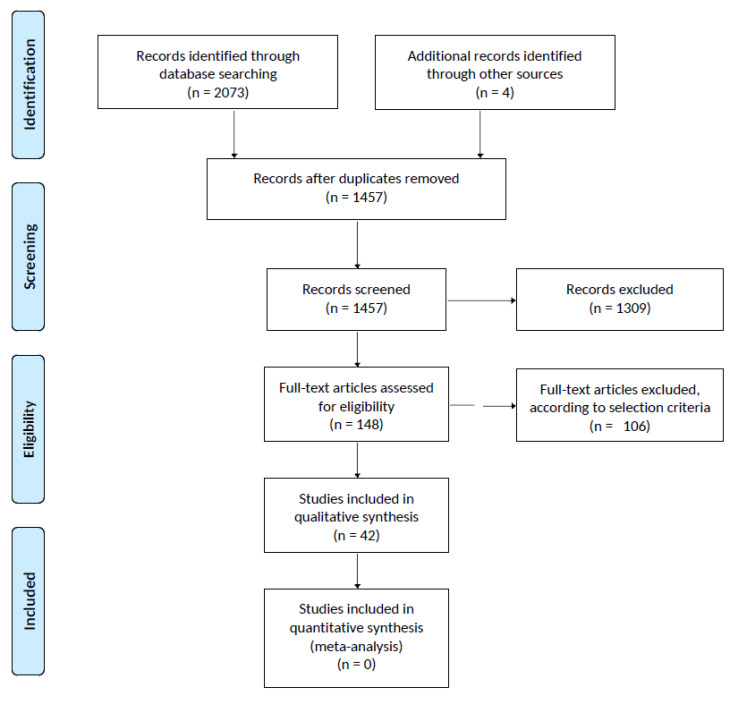
Prisma Flowchart of Review Results.

**Table 1 ijerph-17-03507-t001:** Studies included and group membership based on age-sampling strategies.

Reference	Group
A	B	C	D
Ahn & Jun, 2007 [52]			V	
Aktas et al., 2018 [56]			V	
Ancora et al., 2009 [62]				V
Badr et al., 2010 [64]				V
Bartocci, Bergqvist, Lagercrantz & Anand, 2006 [29]			V	V
Craig, Whitfield, Grunau, Linton & Hadjistavropoulos, 1993 [48]			V	
Evans, McCartney, Lawhon & Galloway, 2005 [14]		V	V	
Fabrizi et al., 2011 [65]				V
Gibbins, Stevens, Beyene et al., 2008 [54]			V	
Gibbins, Stevens, McGrath et al., 2008 [55]			V	
Gibbins & Stevens, 2003 [50]			V	
Gladman & Chiswick, 1990 [59]				V
Goubet, Clifton & Shah, 2001 [43]		V		
Green et al., 2019 [66]				V
Grunau et al., 2005 [34]	V			
Grunau, Oberlander, Whitfield, Fitzgerald & Lee, 2001 [31]	V			
Holsti, Grunau, Oberlander & Whitfield, 2004 [35]	V			
Holsti, Grunau, Oberlander & Whitfield, 2005 [36]	V			
Holsti, Grunau, Oberlander, Whitfield & Weinberg, 2005 [37]	V			
Holsti, Grunau, Whifield, Oberlander & Lindh, 2006 [38]	V			
Holsti, Weinberg, Whitfield & Grunau, 2007 [39]	V			
Johnston, Stevens, Yang & Horton, 1996 [42]		V		
Johnston & Stevens, 1996 [27]	V			
Johnston, Stevens, Yang & Horton, 1995 [60]				V
Klug et al., 2000 [61]				V
Lucas-Thompson et al., 2008 [15]	V	V	V	
Maimon et al., 2013 [40]	V	V	V	
Mehta, Mansfield and VanderVeen, 2010 [46]		V		
Mörelius, Hellström-Westas, Carlén, Norman and Nelson, 2006 [51]			V	
Morison et al., 2003 [41]	V			
Munsters, Wallström, Agren, Norsted and Sindelar, 2012 [47]		V		
Porter, Wolf and Miller, 1999 [16]	V	V	V	
Pourashoori et al., 2018 [57]			V	
Serpa et al., 2007 [45]		V		
Schenk et al., 2019 [58]			V	
Slater et al., 2006 [26]				V
Slater et al., 2009 [63]				V
Slater et al., 2010 [28]	V			
Stevens et al., 2007 [53]			V	
Walden et al., 2001 [44]		V		
Williams, Khattak, Garza and Lasky, 2009 [30]		V	V	
Xia et al., 2002 [49]			V

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
