# Peer review of "Methodological Issues in the Study of the Development of Pain Responsivity in Preterm Neonates: A Systematic Review"

_ijerph, 2020, doi:10.3390/ijerph17103507_

Round 1

Reviewer 1 Report

Methodological issues in the study of the development of pain responsivity in preterm neonates: A systematic review

The author presents a systemic review with a central theme on the impact of age variables (GA, PMA, and CH) used in several studies in the interpretation of pain responsivity. The author proposes innovative approaches and methodological consistency for accurate analysis of pain responsivity. The manuscript is overall well written but needs some clarifications.

My comments and questions to the authors are as below.

  1. As the study deals with the preterm babies, the choice of 45 weeks PMA as selection criteria is not defended. Was this decided a priori, and if so, on what basis. If this was just an exploration of different timescales, then this should be stated.
  2. The flow chart and checklist are nicely done, but the authors should consider adding a table summarizing/categorizing 42 studies that have been reported, which would be helpful.
  3. The authors should consider analyzing the inter-study variability (heterogeneity) in outcomes of different groups of studies, and the authors should also highlight the proportion of patient cohorts with and without controls. 
  4. The authors should elaborate more on the interesting “age sampling strategies” describe in this review that was more accurate in the interpretation of the pain responsivity.
  5. With substantial variations in outcome measures and methodology among the studies, how does the author explain the general trend showing “GA and PMA related increase in facial pain responses with more immature neonates showing dampened responsivity?
  6. With so many variabilities illustrated through results in age terminology, GA assessment methods, outcome measures, and age-specific strategies, the authors should discuss and elaborate on their recommendations in the discussion for accurate interpretation of pain responsivity.

Reviewer 2 Report

The topic of the manuscript “Methodological issues in the study of the development of pain responsivity in preterm neonates: A systematic review” is very interesting issue for the IJERPH readers. However, the authors should provide additional information in order to allow the reader to better understand the interest of the study and its results. Likewise, the information is given following an organization that is not aligned with the PRISMA statement. Consequently, it makes difficult to understand the content.

a) Abstract: The summary structure is not clear. Additionally, according to the PRISMA statement, some background related to the main topic must be included.

b) Introduction: The authors should avoid the use of lists. Instead the information shall be provided throughout the text (lines 55-61).

c) Introduction: The description of prematurity or preterm birth, a key concept in the work, is not described in the introduction. The description of this term, together with its classifications following the recommendations of the World Health Organization (WHO), must be included.

d) Introduction: The queries raised with the study must be explicitly stated in the objectives epigraph. It is advisable to review this section given that is slightly confusing. Likewise, the title of subsection 1.5 must be reformulated (line 103).

e) Methods: The PRISMA statement introduces the best structure to follow in the “Methods” section, to organize the information in a way that facilitates the readers understanding. It is highly recommendable to follow these recommendations. For instance, the "eligibility criteria" (line 126) must be provided before the "search strategy" (line 122).

f) Methods: More detail about the sources of information must be included, such as the search periods and the filters used.

g) Methods: The information provided about the selection criteria used for the study is insufficient (line 125). More details should be given, and the content must be reorganized according to the PRISMA statement.

j) Results: The flow diagram depicted in figure 1 (essential element in a systematic review) must be explicitly referenced in the first paragraph of this section. This would enable the readers to easily consult this essential information.

k) Conclusions: This epigraph must only contain a general interpretation of the results as well as provide evidence about possible future implications of this study, both at clinical and research level. The “Conclusions” section must be reviewed since it contains a lot of information that can be incorporated in previous sections.

l) References: The bibliographic references must be reviewed. The Vancouver style does not include the “doi” as an element in the general outline of scientific articles.

Reviewer 3 Report

Important and thorough systematic review of the neonatal pain literature.  Did not see heart rate variability(HRV) mentioned, a well-documented and now robust way of tracking autonomic nervous system function in real time.  Could you comment in your discussion on this omission and your thoughts about including HRV in future studies?

Round 2

Reviewer 1 Report

Rebuttal responses are nicely stated. No further comment.